# Towards robust medical machine olfaction: Debiasing GC-MS data enhances prostate cancer diagnosis from urine volatiles

Adan Rotteveel[1◉], Wen-Yee Lee[2◉], Zoi Kountouri[3◉], Nikolas Stefanou[3], Howard Kivell[3,4], Clifford Gluck[3,5], Shuguang Zhang[3,6], Andreas Mershin[3,7,8*]

1 Hcyon Technology, Amsterdam, North Holland, The Netherlands, 2 Department of Chemistry and Biochemistry, University of Texas El Paso, El Paso, Texas, United States of America, 3 RealNose Inc., Arlington, Massachusetts, United States of America, 4 Endless Frontiers Laboratory, New York University Stern School, New York, New York, United States of America, 5 Gluck Urology Clinic and Wellness Center, Hingham, Massachusetts, United States of America, 6 MIT Media Lab, Massachusetts Institute of Technology, Cambridge, Massachusetts, United States of America, 7 Osmocosm Foundation, Boston, Massachusetts, United States of America, 8 MIT Sloan School of Management, Massachusetts Institute of Technology, Cambridge, Massachusetts, United States of America

◉ These authors contributed equally to this work.

* mershin@mit.edu

**Data availability statement:** All GC-MS files are available from the Kaggle website at the url: https://www.kaggle.com/datasets/adanrott/ prostate-cancer-gc-ms-data.

## Abstract

Prostate cancer (PCa) is a major, and increasingly global, health concern with current screening and diagnostic tools' severe limitations causing unnecessary, invasive biopsy procedures. While gas chromatography–mass spectrometry (GC-MS) has been used to detect urinary volatile organic compounds (VOCs) associated with PCa, efforts to identify consistent molecular biomarkers have failed to generalize across studies. Inspired by the olfactory diagnostic capabilities of medical detection dogs, we do not reduce chromatograms to a list of compounds and concentrations. Instead, we deploy a machine learning approach that bypasses molecular identification: PCa "scent character" signatures are extracted from raw time series data transformed into image representations for classification via convolutional neural networks. To address confounding factors such as sample-source bias, we implement a multi-step pre-processing and debiasing pipeline, including empirical Bayes correction, baseline drift removal, and domain adversarial learning. The resulting model achieves classification performance on par with similarly trained canines, achieving a recall of 88% and an F1-score of 0.78. These findings demonstrate that, at least in the context of PCa detection from urine, machine learning-based scent signature analysis can serve as a fully non-invasive diagnostic alternative, with these early results being also relevant to the wider emergent field of medical machine olfaction.

**Funding:** This work was supported in part by RealNose Inc., a startup specializing in machine olfaction (www.realnose.ai). Authors A.R. (contractor), C.G., N.S., A.M., H.K., S.Z., and Z.K. are affiliated with RealNose Inc. RealNose Inc. had no role in study design, data collection and analysis, decision to publish, or preparation of the manuscript.

**Competing interests:** RealNose.ai develops machine olfaction technologies, and the findings of this research on de-biasing may have implications for future product development. A provisional patent application related to this work has been filed (Provisional Patent No. US 63/659,630). These interests have been fully disclosed. RealNose.ai and all affiliated co-authors affirm that these do not alter our adherence to PLOS ONE policies on sharing data and materials.

## Introduction

Prostate cancer (PCa) is the second most commonly diagnosed cancer in men worldwide [1]. In 2022 alone, PCa accounted for 1.4 million new cases, leading to 397,000 deaths globally [1, 2]. The disease burden is projected to increase substantially, with incidence rates expected to rise to 2.9 million by 2040, with 700,000 dead annually, representing an 85% increase over 20 years (see also the erratum for a figure correction) [3,4]. The prevalence of PCa is strongly age-dependent, with studies estimating that 59% of men over 79 years old have been affected by the disease [5].

Despite widespread screening efforts, current diagnostic modalities suffer from well-documented limitations. The prostate-specific antigen (PSA) test, although widely used, lacks tumor specificity and exhibits a low specificity of approximately 20%, leading to a high false positive rate and unnecessary biopsies [6–8]. Digital rectal examination (DRE) has a reported sensitivity of 51% and specificity of 59%, making it an unreliable standalone diagnostic tool [9]. While costly, multiparametric magnetic resonance imaging (mpMRI) has improved detection rates, achieving a sensitivity of 93%, its specificity remains suboptimal at 41%, often requiring confirmatory biopsies [10]. The 2017 PROMIS study highlighted these limitations, reporting that among 158 patients who tested negative via MRI, 17 were later found to have clinically significant cancer upon biopsy [10].

We here examine a potentially more reliable, non-invasive alternative method leveraging volatile organic compound (VOC) profiling in the headspace of biological fluids such as urine. Medical detection dogs trained to detect PCa from urine via olfaction have demonstrated remarkable sensitivity and specificity, with some studies reporting rates as high as 98.7% or even 100% [11]. This phenomenon, especially when seen in the context of decades of published work about the diagnostic and tracking abilities of many animals, suggests that unique perceptual signatures (recognizable "scent characters") conferred by VOCs may serve as reliable, stable biomarkers for PCa diagnosis [12].

However, attempts to capture the salient signal as a list of molecules using analytical frameworks such as GC-MS have faced challenges in generalizability and practical utility. These difficulties stem primarily from variability in identified VOC identities across studies, inconsistent biomarker library standards, and confounding batch effects arising from differences in sample collection, processing, and instrument calibration, among other factors [13–16].

To address some of these limitations, while preserving existing GC-MS datasets in their rawest form (in order to not lose information) we developed a novel machine olfaction approach that moves beyond conventional reduction of the signal into a list of VOC identifications. We instead leverage deep learning techniques to classify *scent character* patterns directly from GC-MS ion chromatograms. We do so by transforming raw GC-MS time series data into two-dimensional image representations, which are then processed by convolutional neural networks (CNNs) trained on well-established machine vision principles. Furthermore, to mitigate the confounding effects of sample source bias, we implement a robust debiasing pipeline that incorporates empirical Bayes correction, baseline drift removal, and domain adversarial learning.

Our study is based on a dataset of 387 urine samples collected from four medical centers across the United States. After excluding corrupted data, 365 samples were used, comprising 125 control cases (biopsy-negative, including 18 benign prostatic hyperplasia cases), 133 low-risk PCa cases (Gleason score 6 and below), and 107 high-risk PCa cases (Gleason scores 7–9). Principal component analysis (PCA) revealed significant clustering of samples based on collection site rather than disease status, underscoring the necessity of implementing bias correction methods prior to classification.

By applying deep learning-based scent characterization, our model achieves a classification accuracy of 75% for distinguishing PCa-positive from PCa-negative cases under five-fold cross-validation. The model demonstrates strong (compared to similarly trained dogs) performance in detecting PCa-positive samples, with a recall of 88% and an F1-score of 0.78, though challenges remain in differentiating between low-risk and high-risk cases. Our results suggest that especially when performance starts approaching that of the best diagnostic dogs, machine olfaction methods will have the potential to provide a scalable and non-invasive, screening and diagnostic tool for early PCa detection and likely other diseases that leave a discernible odor character. We introduce a framework to improve diagnostic reliability in VOC-based (prostate) cancer detection, with broader implications for scent-based and machine olfaction diagnostics becoming available as mainstream clinical medicine.

## Literature review

Studies on using human body fluids such as sweat [17], saliva [18], and breath [19] to detect prostate cancer have been gathering interest. In a 2014 study by Taverna *et al.*, dogs were able to detect PCa in human urine through olfaction [11]. The researchers achieved this by training the dogs to indicate the presence of cancer-specific odor characters in urine samples, conferred to their olfactory apparatus via wafting VOCs emanating from the surface of liquid urine samples at standard temperature and pressure [20]. The dogs presented high rates of sensitivity and specificity (dog 1: sensitivity 100%, specificity 98.7%. dog 2: sensitivity 98.6 %, specificity 97.6 %) [11]. In an effort to mimic the dog's diagnostic capabilities, researchers have been using Gas Chromatography-Mass Spectrometry (GC-MS) to obtain lists of urinary VOCs as potential biomarkers for cancer detection [14], [21], as well as used without first reducing to a list of molecular constituents by two of us (AM and WYL) [13]. Diagnostic-trained animals such as rats, worms, bees, and dogs have outperformed clinical tests and they did so without access to a list of constituent volatiles [12]. The integration of canine olfaction with chemical profiling has shown promise in detecting lethal prostate cancer through urinary VOC analysis trained on the same urine samples as dogs [13]. Additionally, a systematic review and meta-analysis on the olfactory ability of medical detection canines to identify prostate cancer from urine samples further supports the potential of VOC profiling in non-invasive cancer detection [22]. Regarding past applications of CNN to Mass Spectrometry, deep learning techniques, such as CNNs, have significantly advanced the analysis of mass spectrometry data. For instance, a deep neural network-based platform, MSpectraAI, was introduced in order to analyze proteome profiles from mass spectrometry data across multiple tumor types, achieving high prediction accuracy [23]. Furthermore, a self-supervised clustering approach using contrastive learning was developed to analyze mass spectrometry imaging data, effectively identifying molecular co-localizations without manual annotations and enhancing the understanding of biochemical pathways [24]. These studies collectively underscore the advancements in "machine olfaction-adjacent" methods such as VOC biomarker research, and the application of CNNs in mass spectrometry while not new it is done here with novel results.

Currently, one of the key limitations of VOC-based diagnosis is the lack of consensus regarding which VOCs serve as reliable biomarkers for any condition such as prostate cancer. Various studies have reported that specific VOC profiles correlate with cancer in their datasets. However, these VOC lists vary dramatically between studies and often do not generalize from one study to another [13–16].

It is important to consider this fact here: trained canines never answer the question of "What is *in* this sample, by molecule name and concentration ?" they only answer "What

does this sample *smell of ?*" Olfactory perception, while clearly *conferred by,* and using receptors *triggering on,* VOCs, does not stop at analytical chemistry space but also operates in *perceived scent character space.* In this paper, we focus on the concept of a prostate cancer biomarker as an emergent scent character and thus propose a pipeline enabling direct classification on GC-MS data without having a list of molecular IDs in the middle. This approach formalizes machine olfactory perception within a structured framework, representing scent character as part of a Synesthetic Memory Object (SMO), *i.e.* a perceptual fingerprint that encodes scent beyond a list of its' molecular components and their measured concentrations. By leveraging this methodology, as detailed in Rotteveel *et al.* [25], we examine what can be accomplished by moving beyond traditional VOC-based biomarker identification towards a more holistic model of scent-based diagnostics that recognizes that the same scent character (such as that of prostate cancer) is detected by dogs in urine samples presenting vastly different compositions of VOCs whose identities are affected by daily diet, yet the signature scent of cancer detected by dogs persists and recognition of it survives the changes in the urine volatilome.

## Materials and methods

### Dataset characteristics

Original data were generated in our research laboratory at the University of Texas El Paso and collected for ongoing research in PCa diagnostics [26]. This dataset included a total of 387 urine samples and was collected from four different medical centers in the United States of America, namely, Duke University Medical Center, Durham, North Carolina; Eastern Virginia Medical Center, Norfolk, Virginia; Michael H. Annabi Internal Medicine Clinic, El Paso, Texas, and Massachusetts General Hospital, Boston, Massachusetts. GC-MS ion chromatograms were obtained but only 365 patients' samples were selected for use after we rejected data that was corrupted. The dataset includes urine samples from patients aged 44-93 who initially showed abnormal PSA results of 4.0 ng/ml or higher, or were found to have abnormalities during a DRE and subsequently underwent a biopsy for further diagnosis. Based on the biopsy result, the dataset is split into three categories; High-Risk (Gleason scores 7 and higher), Low-Risk (Gleason group 6), and Control (prostate biopsy negative, including BPH samples) (S1 Table). For further information on the urine sample collection and the way these were processed for GC-MS analysis, please refer to S1 Appendix.

### Methods overview

GC-MS ion chromatograms are typically presented as time-resolved amplitudes before being converted to a list of molecules. Algorithms find peaks in the data indicative of a molecule and then match its mass per charge ratio (m/z) fingerprint and retention times against one of the various established reference libraries [27].

We note that molecule(s) of interest might not yet be in a library or have overlapping retention times, potentially leaving compounds unidentified, and we see this as an underappreciated problem when reducing a rich raw dataset into a highly information-reducing list of human-readable molecule names and concentrations [28].

Here we depart from the concept of biomarkers as a set of molecular IDs and their concentrations and instead generalize it to mean an arbitrary dimensional fingerprint (of peak amplitudes or receptor activations or anything that can resemble a time series), where an emergent "scent character" can be computed and assigned and whose detection improves with training. To visualize the dataset we operate on here, the raw GC-MS signals were minimally processed

into a Total Ion Chromatogram (TIC) generating a one-dimensional time series of total ion intensity over time. Typical such traces from healthy and prostate cancer positive (as determined via biopsy) are shown in Fig 1A and Fig 1B respectively. To visualize the data, principal component analysis was performed on the TIC data Fig 1C.

Recognizing the significant loss of information while converting the GC-MS data to one-dimensional TICs, we decided to transform the data into images instead. Converting the GC-MS data into images has benefits, because we can then use the spatial dimensions of an image effectively: one axis for representing the m/z ratios, the second axis for representing time (minutes), and a third dimension or color to represent intensity levels in terms of ion count. An additional advantage of working in visual space is that substantial research has been conducted on image processing and computer vision. For example, pre-trained computer vision models such as ResNet have shown good results in scientific domains when fine-tuned appropriately [29]. Computer vision models like ResNet work best with 256 × 256 pixel resolution images, where each pixel contains three color channels, each encoded in 8 bits. Given that the

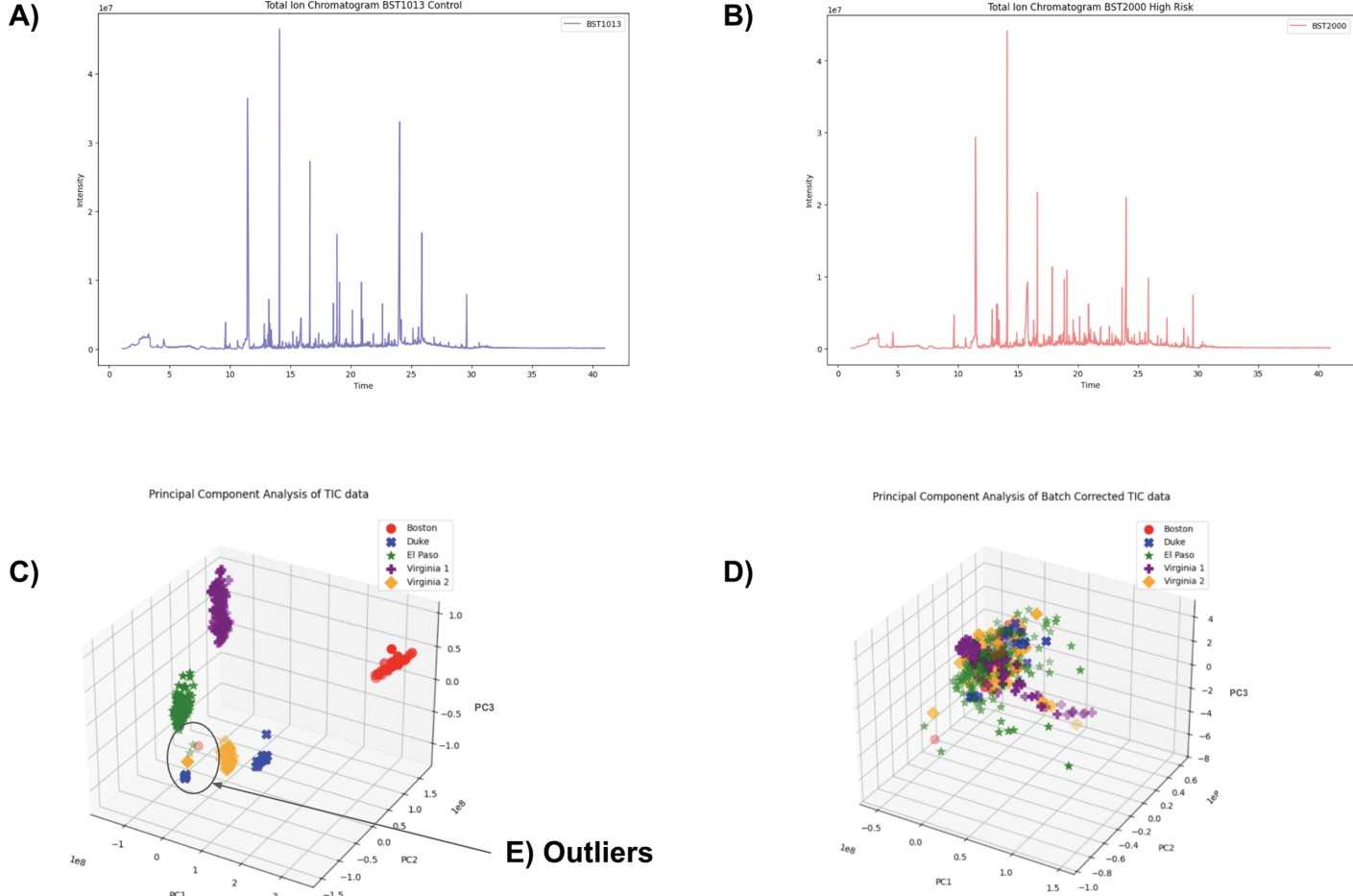

**Fig 1. Time series representing the total ion count over time.** A: Raw total ion chromatograph obtained from a patient testing negative for prostate cancer via biopsy. B: Raw total ion chromatogram obtained from a patient testing positive for prostate cancer via biopsy, Gleason 9 (4+5). C: Principal component analysis shows the data splitting up into source clusters. D: Upon removing the bias using the Empirical Bayes method, the data no longer clusters around the source. E: The outliers come from patients EPPCa 524 (green), EPPCa 51 (green), BST875 (red), BR204_20 (orange), DK4676 (blue), DK2068 (blue), DK4186 (blue), DK4688 (blue) and DK3674 (blue).

GC-MS data originally contains approximately $3.5 \times 10^6$ integer values, the information lost results in a final compression ratio of:

$$\text{Compression Ratio} = \frac{\text{GC-MS Data Size in Bits}}{\text{Image Size in Bits}} = \frac{3.5 \times 10^6 \times 32}{256 \times 256 \times 3 \times 8} \approx 71.2$$

To ensure no important information was lost, we decided to create a machine learning pipeline capable of analyzing "zoomed in" images, as in Fig 2B.

When examining the 3D peak data image closely, numerous small peaks become apparent that are not visible in a zoomed-out view, such as in Fig 2A and in Fig 2D, an alternative heatmap representation of the same data. At the same time, significant noise patterns become evident, such as steady signals at certain m/z values that persist for several minutes before disappearing (Fig 2B). These signals are unlikely to be true peaks and may be the cumulative result of device settings, column bleed, or other artifacts. The pipeline introduced in the next section explains the methods used to eliminate background noise prevalent in the data.

## Preprocessing pipeline

Each sample originally contains approximately 7300 timesteps, and for each time step, there are 480 m/z recordings. Therefore, the first step is to efficiently transform the 480-dimensional time series into a simpler, 1-dimensional time series by computing the Total Ion Chromatogram (TIC). This process involves summing each timestep over all m/z scores, thus creating a 1-dimensional time series representing the total ion count over time, as can be seen in Fig 1A and Fig 1B.

### 1. Rounding m/z values

Firstly, the raw m/z data included non-integer values (e.g., 23.6 and 23.7). To prepare the data for subsequent processing, where each specific m/z ratio's noise patterns are processed differently, the presence of too many different variables due to non-integer values would lead to computational overload. Therefore, each m/z value is rounded to the nearest integer using:

$$m = \text{round}(\text{m/z}_{\text{raw}}) \tag{1}$$

### 2. Segmenting time frames

The data was divided into smaller time frames of $\Delta T = 1.5$, resulting in $N = 28$ segments per $T_{\text{total}} = 41$ minute sample. This segmentation allows the noise and baseline removal algorithms to be applied locally, meaning that the risk of miscategorizing a small peak as noise is significantly reduced.

$$\Delta T = \frac{T_{\text{total}}}{N} = \frac{41 \text{ minutes}}{28} \approx 1.5 \text{ minutes} \tag{2}$$

### 3. Quantile-based noise correction

Within each time frame $T_i$ and for each specific m/z ratio $m$, we calculated the lowest 15th percentile of signal intensity. This quantile, $Q_{15\%}(T_i, m)$, represents a threshold below which the signal is considered baseline noise. It aims at only retaining values that are significantly higher than routine background fluctuations, eliminating insignificant peaks:

$$Q_{15\%}(T_i, m) = \text{Quantile}_{15\%}(\{I_{i,m,k}\}) \tag{3}$$

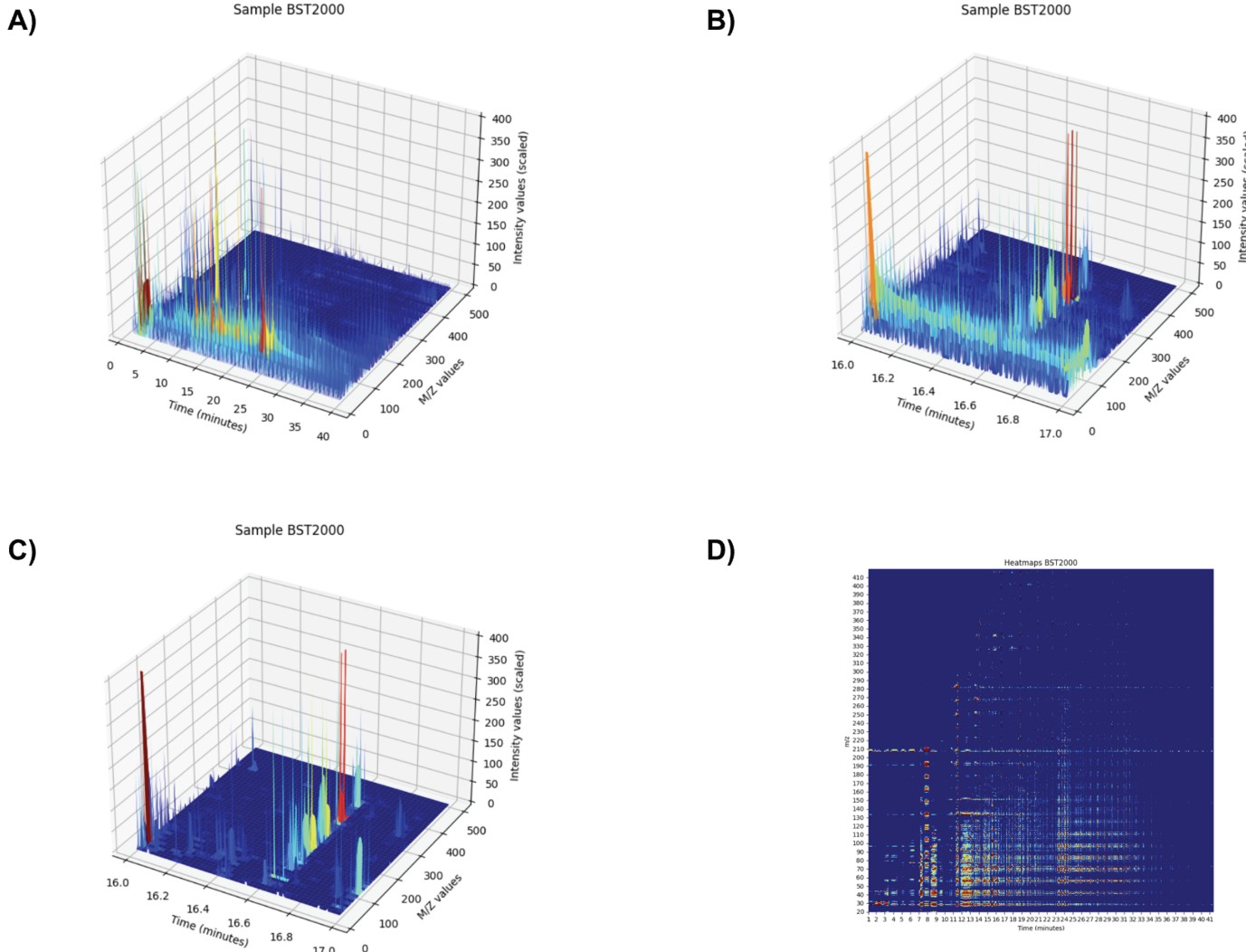

**Fig 2. 3D peaks of patient BST2000.** A: 3D peaks of patient BST2000 shown for a full timeframe of 41.33 minutes. B: 3D peaks of patient BST2000 zoomed in between minutes 16.0 and 17.0. Noise and small peaks show. C: Same timeframe as B but after applying the correction algorithms. Noise is removed and hidden peaks ow show up. D: An alternative visualization of patient BST2000's peak data by converting it to a heatmap.

where $\{I_{i,m,k}\}$ denotes the set of intensity values for m/z $m$ within time frame $T_i$. The baseline noise threshold $Q_{15\%}(T_i, m)$ is then subtracted from all intensity values $I(t,m)$ within the corresponding time frame:

$$I_{\mathrm{corr}}(t,m) = \max\left(I(t,m) - Q_{15\%}(T_i, m),\ 0\right) \tag{4}$$

For example, if $Q_{15\%}(T_i, m) = 400$ ions, then 400 is subtracted from each $I(t,m)$ in that time frame. Applying this technique conserves peaks' position relative to each other and effectively removes noise, as can be seen in Fig 2C.

### 4. Polynomial baseline drift correction

After quantile-based correction, there may still be residual baseline drift across the time frames (Fig 3A). To help remove this drift, a polynomial baseline drift correction algorithm was applied from the Python Baseline Removal library [30,31]. This algorithm was made specifically for spectroscopy data and requires fitting a two-dimensional polynomial to the corrected intensity data $I_{corr}(t, m)$ and then subtracting this fitted baseline from the data to obtain the final processed, baseline drift-free, signal [32] (Fig 3B).

### 5. Mirex normalization

An internal standard is a compound, in this case, Mirex (99.0% purity; Dr. Ehrenstorfer GmbH, Augsburg, Germany), that is added at a constant concentration to all samples. Its consistent presence allows for the correction of variations due to differences in injection volume, instrument sensitivity, or sample processing. By scaling each sample's peak intensities relative to the Mirex peak, this method minimizes amplitude variability. For each sample, the peak area of Mirex is first quantified and then used to adjust the intensities of all other peaks. An example of this process is highlighted in Fig S2C, where the upper panel shows the raw chromatogram with variable Mirex peaks, and the lower panel demonstrates the standardized peaks after normalization.

### Machine learning

To leverage established machine vision techniques, we transformed the GC-MS data into image representations suitable for input into pre-trained CNNs such as ResNet18. In doing so, two main image formats were considered: 3D heat maps, which visually encode the intensity variations across time and m/z values, and 3D peak representations, which emphasize the discrete peak structures in the data (Fig 2C and 2D).

Although GC-MS data are fundamentally a time series of ion intensities across m/z values, converting them into two-dimensional images allows us to exploit the feature extraction capabilities inherent in CNNs. This transformation is not intended to provide a detailed molecular

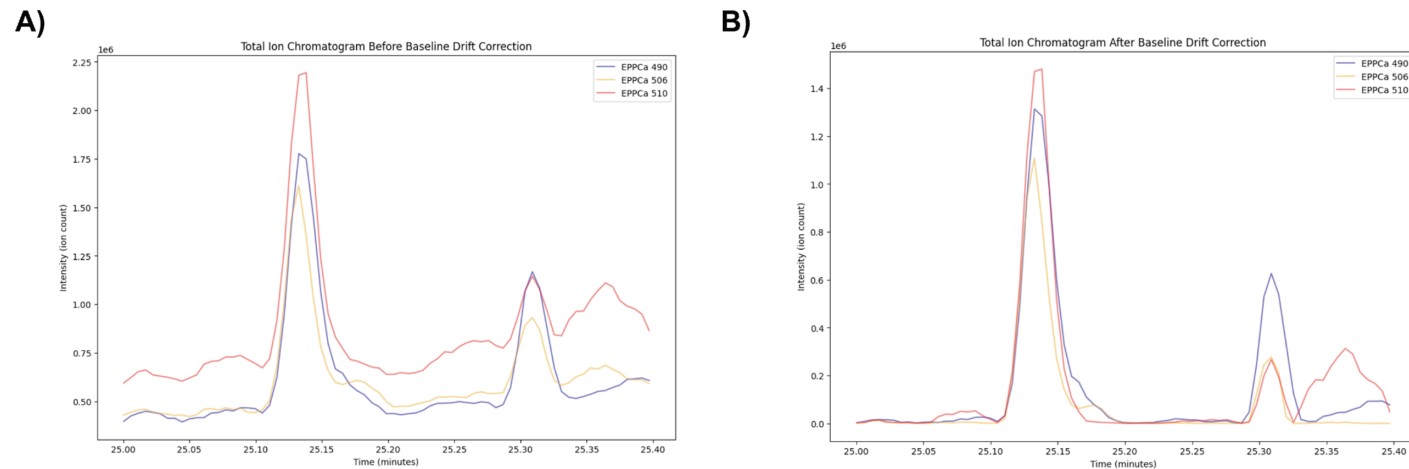

**Fig 3. Total Ion Chromatograms before and after baseline drift correction.** A: Total Ion Chromatograms from patient samples EPPCa 490, EPPCa 506, and EPPCa 510 before baseline correction, captured between 25 and 25.4 minutes. B: Total Ion Chromatograms from patient samples EPPCa 490, EPPCa 506, and EPPCa 510 after applying baseline correction, captured between 25 and 25.4 minutes.

map, but rather to capture complex structural patterns, such as peak shapes, distributions, and spatial relationships that together form parts of an emergent "scent character". ResNet18 is used not to identify individual molecules but to learn high-level representations of the data. This approach is analogous to how human perception of color, shape or sound is based mostly on receptor activation patterns and only rarely on isolated detection events. Even if different molecular compositions yield similar m/z distributions, the emergent patterns captured by the CNN remain uniquely diagnostic. In essence, the ResNet18-based model extracts a robust, high-dimensional signature from the GC-MS data that mirrors the complexity of biological olfaction, much like how trained detection dogs recognize a disease state from a complex scent signature without relying on a predefined list of molecules.

As can be seen in Fig 1D, samples from the same source hospital tended to co-cluster. To counter these source batch effects, we employed the adaptive empirical Bayes algorithm introduced by Johnson *et al.* [33] for microarray data, which we extended to our GC-MS setting. In this algorithm, the prior distribution is inferred directly from the observed data rather than from prior assumptions, adapting to each source. Specifically, we modeled the multiplicative batch effects with a gamma prior $G(\alpha, \beta)$, estimating $\alpha$ and $\beta$ via maximum likelihood, while additive (mean) shifts were captured by a normal prior $N(\mu, \sigma^2)$ whose parameters were likewise inferred from the data. This approach assumes that batch effects remain relatively stable across time points, making it well-suited for our GC-MS dataset.

To reduce source bias further in our predictive model, we integrated a bias removal pipeline into our machine learning framework [34], [35]. Signals transformed into images were used as input to a ResNet18 convolutional neural network, which then compressed them into a latent space [36]. On top of this latent space, two models were built: one aimed at diagnosing the disease and another at predicting the source hospital of the sample [37]. The first model on top of the latent space transforms the latent space such that it is minimally possible to detect source bias using gradient reversal techniques, and the second model, in parallel, transforms the latent space to diagnose the patient. The complete debiasing process is detailed in pseudocode in (S1 Algorithm).

To illustrate our classification pipeline from raw GC-MS data to final classification, we provide an overview of the workflow in Fig 4.

## Results

Upon conducting principal component analysis on the TIC data, we were surprised to observe the data clustered neatly by their source hospital, rather than prostate cancer risk level, as illustrated in Fig 1C. This is a serious problem and we now find to be a pervasive and under-reported issue with many AI diagnostics, as many models end up using predicted socioeconomic status, race and other non-medical parameters that create the illusion of accuracy [38–41]. As shown in Fig 1C, it was almost always the case that data from the same source naturally converged, with only minimal outlying data points (Fig 1C). These outliers exhibited certain special characteristics, such as all outliers having African American or Hispanic heritages, while these groups represent 55% of the total dataset. Furthermore, six of these outliers have high PSA levels, with two showing exceptionally high values (above the 97th percentile). All samples had lower total ion counts than their batch counterparts. Following batch normalization, the mean ion count was 5.197 ($\pm$1.165 SD), with none of the outliers exceeding it (S2 Table).

To counter these source batch effects, we employed the adaptive empirical Bayes algorithm introduced by Johnson *et al.* [33]. Notably, after applying the correction, a simple *k*-means

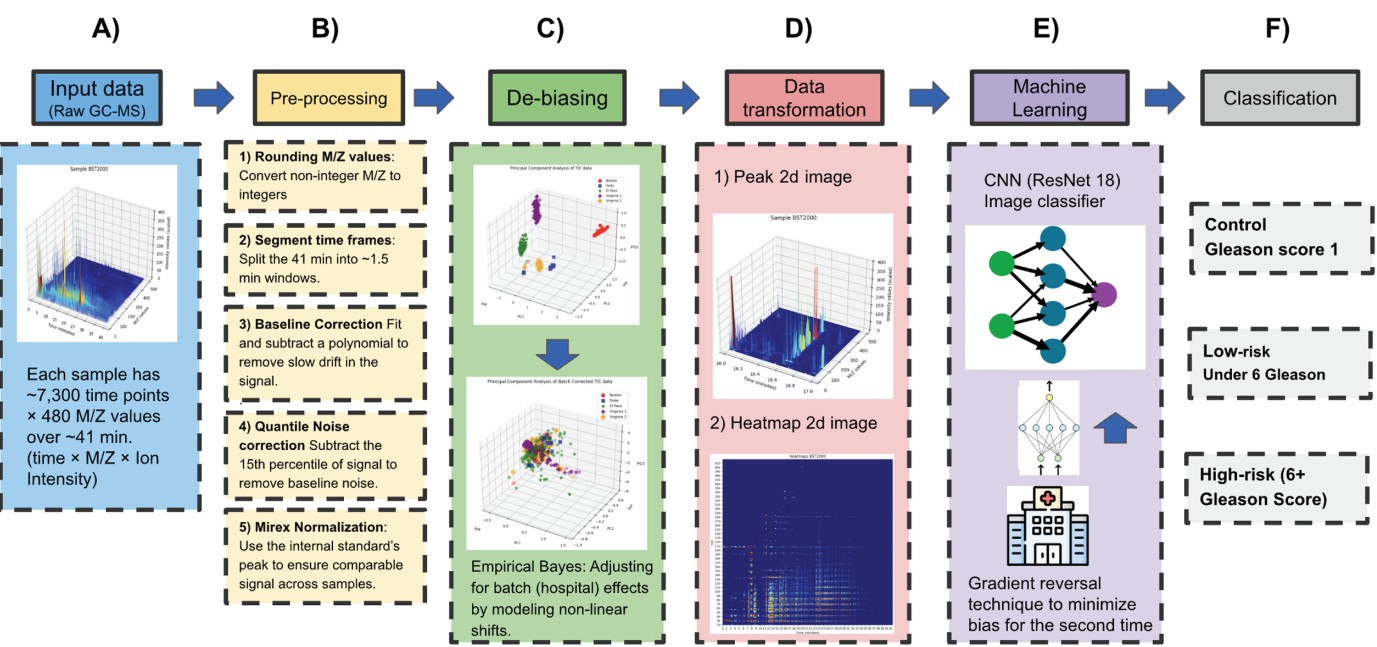

**Fig 4. Full processing pipeline.** A: Each urine sample is measured over ∼ 41 minutes across 480 m/z values, yielding a 3D signal (time, m/z, ion intensity). B: Each sample undergoes pre-processing which includes rounding m/z values, segmenting time frames, baseline drift correction, quantile-based noise correction, and Mirex normalization to ensure comparability across samples. C: An empirical Bayes approach adjusts for source (hospital) batch effects so that downstream classification is not confounded by location. D: Corrected signals are converted into 2D images (e.g., heat maps or 3D-peak projections) for input into convolutional neural networks. E: A deep neural network (e.g., ResNet18) extracts latent features, and a domain-adversarial branch helps remove residual source bias. F: Classification, The model outputs the prostate cancer risk group (High-Risk, Low-Risk, or Control) based on the learned "scent signature."

clustering on source labels dropped from about 96% accuracy to just 27%, indicating substantial bias reduction. We did observe, however, that more sophisticated neural networks trained specifically to predict the source could still detect it above chance, implying a residual signal.

Therefore, to eliminate source bias further from our image data, we integrated a bias removal pipeline into our machine learning framework. This pipeline was thoroughly tested and resulted in lower cross-validated accuracy scores, of 75 percent accuracy, than the original model which was achieving a cross-validated accuracy score of 81 percent. The original model, achieving an accuracy score of 81 percent, did not include the bias removal part in its pipeline, indicating that the model is likely learning partly from the source bias and not the underlying prostate cancer signature conferred in the volatile-dependent GC-MS data.

While our pre-processing methods improved peak detection within individual batches, the identified peaks did not generalize well across different medical centers. For example, certain peak patterns at specific m/z ratios and retention times, distinguished PCa positive from PCa negative samples coming from Boston. However, for other source hospitals, such as El Paso, these peaks were either not prevalent or failed to discriminate between PCa-positive or PCa-negative samples. This means a clustering "key" in the form of a list of peaks (or molecules) would not work as a generalizable handle to cleave a dataset into positive and negative for PCa outside of the dataset it was trained on. Our approach appeared to work initially since after a few epochs it was not possible to use the transformed data for recognizing the source hospital. However, if the bias remover is discontinued and the data are then analyzed by CNNs trained for additional epochs to specifically zoom in and detect the source, we could still identify the source hospital with high accuracy scores of 92 percent and above. It, therefore, appears that

there are additional nonlinear traces within the GC-MS data still indicative of the source hospital. We can imagine this method as a tug-of-war between the neural network removing the source data that allows for easy clustering, and the opposing neural network trying to keep it maximally predictive. This tug-of-war can be shown by defining the loss functions for the risk level as $\mathcal{L}_0(\theta)$ and source location as $\mathcal{L}_q(\theta)$. Then the total loss function of the neural network can be written as:

$$\mathcal{L}_{\text{total}}(\theta, \lambda) = \mathcal{L}_0(\theta) - \lambda \mathcal{L}_q(\theta) \tag{5}$$

With $\lambda \in [0,1]$ as the hyperparameter balancing between the maximization of the source prediction loss $\mathcal{L}_q(\theta)$ and minimizing the risk prediction loss $\mathcal{L}_0(\theta)$, during the final minimization of the total loss $\mathcal{L}_{\text{total}}(\theta, \lambda)$. Similarly to the temperature parameter in Large Language Models (LLMs), which controls the level of randomness allowed in predicting the next word, the strength of the neural network in removing source data can be adjusted by modifying the degree of data transformation. This method has the potential for further improvement by giving us the opportunity to adjust the amount of bias removal, such that it does not also remove the usable signal. Furthermore, including the transformation strength variable in a standard neural network hyperparameter search results in an automated process towards finding the correct value for each case [42].

Removing bias reduced our cancer-vs-non-cancer classification accuracy from about 81% (with bias) to 75% (after debiasing), highlighting a trade-off whereby part of the initial "accuracy" originated from exploiting confounding batch information (Fig 1D).

We categorized samples based on their Gleason Scores as follows: biopsy⬚negative samples were designated as Control, samples with a Gleason score of 6 or lower were labeled Low-Risk, and those with scores of 7 or higher were classified as High-Risk. As shown in Fig 5, our three-class classification (Control, Low-Risk, High-Risk) yields moderate performance overall, with a particularly high success rate in distinguishing prostate cancer negative samples from prostate cancer positive samples(Low-Risk + High-Risk). Fig 5A depicts the confusion matrix of total counts, indicating that most control samples are correctly recognized, whereas Low-Risk and High-Risk are misclassified more often, particularly into the category of the other. While our method achieves approximately 75% accuracy in separating "cancer" from "non-cancer" under 5-fold cross-validation (using a stratified 20% test set of around 73 samples per fold), it remains challenging to clearly discriminate Low-Risk from High-Risk cases (Fig 5B). Specifically, for the PCa+ class (Low-Risk + High-Risk), the model attains a recall of 0.88 and precision of 0.70 (F1-score 0.78), whereas the control class yields a recall of 0.62 and precision of 0.83 (F1-score 0.71), leading to an overall macro-averaged F1-score of 0.74. Details on the hyperparameter tuning process and final model configuration are provided in Appendix S2.

## Discussion

### Debiasing

The original impetus behind the work presented here was to build out new synaesthetic methods to diagnose prostate cancer (PCa) from raw data of a variety of types [25]. We chose GC-MS as our first data type and pre-treated it by removing the source bias (Fig 1D) and the systemic noise and baseline drifts. Any training on untreated data would have yielded bias-contaminated results, potentially giving a falsely inflated accuracy and eventually undermining confidence. We find that such biases and systematic errors are common issues in AI-driven cancer detection methods [40,43], so we created the suite of debiasing tools presented here as a pipeline (S2 Fig) [35]. This was a necessary step before we could use the GC-MS data

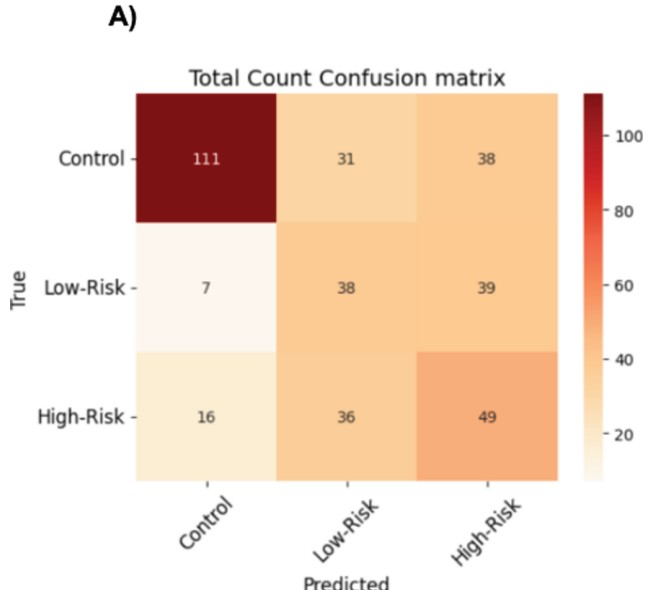

**Fig 5. Confusion Matrix for Three Risk Levels (Control, Low-Risk, High-Risk.** This figure summarizes the model's performance across a 5-fold cross-validation, each fold using a stratified 20% test set (n ≈ 73 test samples per fold). A: The Total Count Confusion Matrix for the three risk groups (Control, Low-Risk, High-Risk). We see that most Control samples are correctly classified (111 out of 180), while Low-Risk and High-Risk cases exhibit notable overlap. B: The Classification Report for a two-class scenario (merging Low-Risk and High-Risk as PCa+). The model achieves an overall accuracy of 75%, with higher recall (0.88) for the PCa+ class but slightly lower precision (0.70). Conversely, Control exhibits lower recall (0.62) but higher precision (0.83). The macro-averaged F1-score of 0.74 indicates reasonably balanced performance across both classes. The performance metrics are defined as follows: *Precision* is defined as the number of true positives (TP) divided by the sum of true positives and false positives (FP); *Recall* is defined as the number of true positives (TP) divided by the sum of true positives and false negatives (FN), and the *F1-score* is computed as 2 times the product of Precision and Recall divided by the sum of Precision and Recall. The term "support" refers to the number of actual instances for each class.

fully and before we could let the classification models learn directly from the data. Note that in no part of the process do we reduce these data to a list of molecules' names and their concentrations. Instead, we transform the data into images, enabling the development of what we call synaesthetic memory objects (SMOs). We realized that to continue developing diagnostic algorithms without first removing the source hospital name bias would lead to error-prone results.

To remove source bias, an empirical Bayes method [33] and a bias removal pipeline were used. Although these methods were successful in removing source bias to a large extent, when tested with advanced vision neural networks we found that we could still detect the source hospital. This revealed a deeply ingrained source bias, which may have occurred due to the way that the urine samples were collected and handled at each hospital. This makes the source effect difficult to completely eliminate and furthers our understanding of how source bias can be addressed, which is especially critical for research on generalizable biomarkers from urine scent. Diagnostic AI developers must be aware of the severity of such biases potentially influencing their results, creating the illusion of a causal, clinically relevant relationship when it does not exist. It is important to carefully monitor the presence of source bias and to use robust bias removal methods to mitigate it while also quantifying the limitations on accuracy caused by residual source bias and other non-medically relevant biases. After applying our pre-processing methods, we were able to identify discriminative peaks within individual batches, but they did not align across different medical centers. This suggests, once again,

that location-specific effects are at play and that due to some systematic, handling, or processing reasons, a consistent pattern across all medical centers cannot be established and instead each medical center shows up as an accurate clustering label (Fig 1C). While manageable through methods such as those we described here, the prevalence of biases in datasets used for predictive models will likely be widespread, influenced by confounding variables such as location, ethnicity, age, and other hard-to-tease-out factors that could affect the final prediction [41]. The presence of confounding factors in the training set is not a problem isolated to machine learning; canines trained on a poorly chosen set of positive (foreground) and negative (background) urine samples can learn to trigger on the wrong signal. Consider a biodetection canine that is asked to detect prostate cancer but all the negative controls are female urine. The clever dog soon starts indicating the targets but is in fact simply choosing the "non-female urine" because that is an easier-to-detect olfactory handle, and in this case, it happens to perfectly correlate with the presence of the target signal. In this thought experiment, based on a real-world case, even though the dog successfully identifies the target among the decoys, it is not a dog trained on prostate cancer, it is a dog trained on finding male urine in a background of female urine. Once such a dog is challenged with decoys from healthy males its sensitivity and specificity can be expected to decrease dramatically. Similar issues arise when AI is trained on biased and improperly controlled datasets (of any type).

## Limitations

Besides the constraint of trace source bias, our study also had another limitation, which appeared due to our need to remove said bias: it is possible the signal – the scent character of PCa – is lost due to over-applying our debiasing process. While we mitigated the loss of information by performing minimal corrections, there is a non-zero chance our methods can end up "cleaning away" the clinically relevant signal too. Moreover, GC-MS as a laboratory technique may miss-identify or completely miss certain trace molecular species and while some machines can theoretically reach limits of detection in the low femtogram ranges under ideal conditions (e.g. such as when using octafluoronaphthalene as a benchmark in a near ideal carrier gas background), when outside those parameters, such as when exposed to actual urine samples which are more complex than any single molecule [44], sensitivity rates of as low as 3-4 parts per trillion can be achieved by advanced techniques that complement conventional GC-MS, such as GC-MS/MS [45]. For the purposes of our dataset, we note that even the internal standard compound Mirex is undetectable in several samples, casting doubt on the reproducibility of trace compound identification in a complex background headspace such as urine. We also note that biological olfactory receptors would likely be able to trigger their cognate odorants when presented in similarly complex backgrounds and variable sample conditions. This is due to the fact that GPCRs are single-molecule detectors and could theoretically trigger a signaling cascade terminating in an olfactory sensory neuron's action potential being fired as a result of a single molecule being present in a sample, but only when the interaction time between sample and receptor is long enough for the odorant to interact with the receptor's binding site. Additionally, we acknowledge that the sample size of this study presents as a limitation, as always with any machine learning exercise, a larger sample size would improve the robustness of our findings and reduce the risk of missing medically relevant patterns.

## Future research directions

Relevant to our goal of olfactory diagnosis of prostate cancer, we note that homologs of human olfactory receptors OR51E1 and OR51E2 are overexpressed in human prostate cancer tumor tissue. The homologs have been named PSGR (Prostate Specific G-Protein Coupled

Receptor) and PSGR2, respectively [46]. The overexpression of PSGR and PSGR2 in cancerous prostate tissue cells in comparison to healthy prostate tissue suggests that these GPCRs play a role in signaling among the proliferating prostate cancer cells [47], [48]. The cognate ligands of these receptors potentially (soluble) odorants are likely also produced by the tumor tissue, using metabolic pathways that are likely modified from their healthy state. While many studies focus exclusively on receptor deorphanization to identify activating molecules, we speculate that molecules alone might not be the only useful layer of abstraction needed here. Since olfactory receptor GPCRs have co-evolved with the specific enzymes responsible for producing the odorants these GPCRs trigger on, we have directed our future research into the relationships between receptors and the metabolic machinery producing their odorant,

**Algorithm 1. Pseudocode debiasing GC-MS Data.**

```
Data: Raw GC-MS signals and corresponding metadata
Result: Final classification label (Control, Low-Risk, or High-Risk)

// Step 1: Data Loading
data ← loadGCMS();
metadata ← loadMetadata();

// Step 2: Data Preprocessing
preprocessed_data ← preprocess_data(data) ;          // Remove noise, correct
 baseline, normalize
;
data_corrected ← removeBias(preprocessed_data) ;     // Johnson et al. (2006)
;
// Step 3: Image Transformation
(images, riskLabels, sourceLabels) ← makeImages(data_corrected, metadata);

// Step 4: Model Initialization
model ← ResNet18_pretrained();

// Helper Functions
predictRiskClass(model, features) return model.riskHead(features) ;
// Predict Control, Low-Risk, or High-Risk

predictSource(model, features) return model.sourceHead(features) ;    // Predict
source/hospital label

// Step 5: Training Loop
for epoch ← 1 to num_epochs do

    foreach batch in getBatches(images, riskLabels, sourceLabels) do
        feats ← model.featureExtractor(batch.images);

        riskOut ← predictRiskClass(model, feats);
        sourceOut ← predictSource(model, feats);

        riskLoss ← crossEntropy(riskOut, batch.riskLabels);
        sourceLoss ← crossEntropy(sourceOut, batch.sourceLabels);

        totalLoss ← riskLoss + sourceLoss;

        model.zeroGrad();
        totalLoss.backward();

        foreach param in model.featureExtractor.parameters() do
            grad_risk ← getRiskGrad(param);
            grad_source ← getSourceGrad(param);
            param.grad ← grad_risk − λ_val × grad_source;

        model.updateParameters();

// Step 6: Inference
new_feats ← model.featureExtractor(new_images);
riskScores ← predictRiskClass(model, new_feats);
finalLabel ← argmax(riskScores);
```

specifically employing machine learning models to predict enzymes related to a given GPCR. We suspect that any relationship between the metabolic machinery involved in producing an odorant and the receptor tuned to that odorant will be detectable at the scales of the primary nucleic and amino acid sequences involved in coding the metabolism and detection of its products and finding such will be a boost to machine olfaction efforts using olfactory receptors.

## Supporting information

**S1 Table. Each medical center's contribution of samples across different risk levels.** Table notes: Each medical center has a different PCa risk distribution; for example, Virginia 1 contains only PCa-positive samples and is therefore highly biased. Overall, 10.1% of the samples come from Massachusetts General Hospital, Boston, Massachusetts, 7.7% come from Duke University Medical Center, Durham, North Carolina, 38.9% come from Michael H. Annabi Internal Medicine Clinic, El Paso, Texas, and 43.3% come from Eastern Virginia Medical Center (batch 1 is 24.4%, sample 2 is 18.9%).
(TIF)

**S2 Table. Outliers identified through principal component analysis of the raw dataset.** The outliers are predominantly from African American and Hispanic heritages, although these groups only represent 55% of the total dataset. Furthermore, six of these outliers have high PSA levels, with two showing exceptionally high values (over the 97th percentile). All samples had lower total ion counts than their batch counterparts. Following batch normalization, the mean ion count was 5.197 ($\pm$1.165 SD), with none of the outliers exceeding it.
(TIF)

**S1 Fig. Total Ion Chromatogram Before and After Mirex Normalization. A)** Total ion chromatogram of 15 high-risk samples from El Paso, between 29.4 and 29.8 minutes, showing the internal standard Mirex compound peaks. Variation in peak areas indicates the need for normalization. **B)** Total ion chromatogram of 15 high-risk samples from El Paso, between 29.4 and 29.8 minutes, after Mirex normalization to ensure sample comparability.
(TIF)

**S2 Fig. Correction Pipeline. A)** Baseline Drift Correction **B)** Quantile Noise Correction **C)** Mirex Normalization **D)** Bias correction (Source Bias Removal).
(TIF)

**S1 Appendix. Dataset characteristics.**
(PDF)

**S2 Appendix. Hyperparameter tuning and model configuration.**
(PDF)

## Acknowledgments

We are thankful to Dr. Claire Guest of Medical Detection Dogs UK for enlightening conversations about biodetection canine training and testing, Katia Katsari of General Electric Corporation for alerting us on the importance of the problem of diagnostic AI bias, Prof. Simmie Foster of the Harvard Laboratory for Hot and Cool Research and Massachusetts General Hospital for ideation and editing, Dr. Howard Soule of the Prostate Cancer Foundation for his support of our prostate cancer machine olfaction diagnostic effort, Ruben Sanders of Erasmus School of Economics, Ms Despoina Gerasoudi and the students of the MIT Sloan Executive

Education Advanced Management and Global CEO Programs and other participants in the "Lab-to-Market the MIT Way" and "Making Sense of Scent" (MIT IAP) classes for early feedback on our findings. We also are grateful for their early and critical support to: Nelson Wang, George Kung of KungHo Fund, and Armando Cañas and Robert Jursich of Mavericus Fund.

## Author contributions

**Conceptualization:** Zoi Kountouri, Andreas Mershin.

**Data curation:** Wen-Yee Lee.

**Formal analysis:** Adan Rotteveel, Zoi Kountouri, Andreas Mershin.

**Funding acquisition:** Andreas Mershin.

**Methodology:** Adan Rotteveel, Andreas Mershin.

**Project administration:** Nikolas Stefanou, Andreas Mershin.

**Resources:** Wen-Yee Lee.

**Software:** Adan Rotteveel, Zoi Kountouri.

**Supervision:** Nikolas Stefanou, Andreas Mershin.

**Validation:** Adan Rotteveel, Andreas Mershin.

**Visualization:** Adan Rotteveel, Zoi Kountouri, Andreas Mershin.

**Writing – original draft:** Adan Rotteveel, Zoi Kountouri, Andreas Mershin.

**Writing – review & editing:** Adan Rotteveel, Wen-Yee Lee, Zoi Kountouri, Nikolas Stefanou, Howard Kivell, Clifford Gluck, Shuguang Zhang, Andreas Mershin.

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
