## [Decision Letter · Decision Letter 0]

14 Feb 2025

PONE-D-24-52113Towards robust machine olfaction: debiasing GC-MS data enhances prostate cancer diagnosis from urine volatilesPLOS ONE

Dear Dr. Rotteveel,

Thank you for submitting your manuscript to PLOS ONE. After careful consideration, we feel that it has merit but does not fully meet PLOS ONE’s publication criteria as it currently stands. Therefore, we invite you to submit a revised version of the manuscript that addresses the points raised during the review process.

We look forward to receiving your revised manuscript.

Kind regards,

Li Yang, M.D.

Academic Editor

PLOS ONE

Journal Requirements:

“startup RealNose.ai has supported part of this project.

Co-authors involved in startup:

A Rotteveel (as a contractor)

C Gluck

N Stefanou

A Mershin

Startup:

RealNose Inc

realnose.ai

The paper was mostly written by startup staff.”

“The authors have read the journal’s policy and have the following competing interests: Co-authors Rotteveel, Stefanou, Gluck and Mershin have a financial interest in startup RealNose.ai which has supported part of this project.”

4. In the online submission form, you indicated that “The data underlying the results presented in the study are available from Prof Wen-Yee Lee (contact via email wylee@utep.edu)”

Reviewers' comments:

Reviewer's Responses to Questions

**Comments to the Author**

1. Is the manuscript technically sound, and do the data support the conclusions?

Reviewer #1: Yes

Reviewer #2: Partly

Reviewer #3: Yes

Reviewer #4: Yes

2. Has the statistical analysis been performed appropriately and rigorously? 

Reviewer #1: Yes

Reviewer #2: Yes

Reviewer #3: Yes

Reviewer #4: Yes

3. Have the authors made all data underlying the findings in their manuscript fully available?

Reviewer #1: Yes

Reviewer #2: No

Reviewer #3: Yes

Reviewer #4: No

4. Is the manuscript presented in an intelligible fashion and written in standard English?

Reviewer #1: Yes

Reviewer #2: No

Reviewer #3: Yes

Reviewer #4: No

5. Review Comments to the Author

Reviewer #1: This is amazing work.

Although, as the limitations rightly acknowledge, the true mimicry of the original concept may be too far off at this point, the attempt itself to apply both MS and ML to olfactory senses in diagnosing PCa is a brilliant idea and fully worth pursuing in the future.

However, I do recommend some additional corrections to improve readability.

1. Workflow of data.

As with most ML papers, a comprehensive workflow of how the data was processed would improve understanding. A diagram depicting the 4th paragraph of the methods would help improve understanding.

2. CNN for MS signals?

My understanding of the methods, thus, may be unclear, but is it correct for me to understand that you have processed the MS signals through Resnes as images?

While GS-MS outputs are, in someway, 2 dimensional, it could hardly be considered an 'image' that should/could be processed through a CNN, as the 2 dimensional array is only something based on charge/size and doesn't convey any comprehensive information of individual identities of the molecules in the array.

For instance, wouldn't a configuration of similarly distributed charge/size molecules, but in an entirely unrelated molecular composition be deemed similar?

TL;DR: Please, help me understand the wisdom behind applying 2 dimensional image analysis to a molecular array.

3. Small sample.

Primarily as a urologic surgeon, I am more curious as to why the authors chose to use such a diverse patient group. Limiting them to a limited Gleason Score group might have homogenized the characteristics better. 400 or so samples are too few to train a model.

Could you offer a reply?

Thank you.

Reviewer #2: This study proposes an innovative approach to enhance machine olfaction for prostate cancer (PCa) diagnosis using gas chromatography-mass spectrometry (GC-MS) data. The authors introduce a novel debiasing method to address source-related biases and employ machine learning techniques to classify urinary samples. The topic is highly relevant, especially given the need for non-invasive diagnostic tools for PCa. The manuscript is well-structured and clearly presents the motivation, methodology, and results.

However, there are several areas where the manuscript can be improved to meet the standards of a high-impact international journal. These areas relate to methodological transparency, scientific rigor, and presentation.

1. The introduction is well-written but could benefit from a stronger emphasis on the novelty of the "scent character" approach

2. Provide more detail on the configuration of the Empirical Bayes debiasing technique, such as the exact distributions used and how they were validated.

3. Elaborate on how hyperparameters for the convolutional neural network (CNN) and ResNet18 were selected. Was there any hyperparameter tuning process?

4. Clarify the rationale for transforming GC-MS data into 3D images instead of other approaches, such as feature extraction or embeddings.

5. Provide access to the Python code or pseudo-code of the debiasing pipeline to ensure reproducibility.

6. Provide a brief explanation of how the “emergent scent character” relates to standard biomarkers and how it advances existing methodologies.

7. Potential data imbalance and demographic bias could influence the model's performance. Quantify and report the impact of these imbalances on model performance using metrics like accuracy, precision, recall, and F1.

8. Provide a confusion matrix and classification report, including sensitivity, specificity, and AUC, for a complete evaluation of model performance.

9. Improve the clarity of figures by adding descriptive legends and using distinct colors for different categories.

Reviewer #3: 1) The abstract is not expressing the novelty of the proposed approach. The whole abstract is not impressive and needs to be rewritten. Should focus on what is problem, why it is important to be solved. How it is solved and what are findings.

2) Introduction section requires reorganization and missing the novelty of the proposed approach, hence, the author is suggested to rewrite the introduction section.

3) The author is requested to include literature review section within the manuscript and here should be some lines of text between the main and sub heading. This rule should be follow in whole paper.

4) Data characteristic should be place in experiments or result section. In the meantime, the methodology section needs the revision and major attention of the author.

5) Results section needs major revision which demonstrates the training as well the testing results. In the meantime the impacts of each consider parameter in a detailed manner which improves the validity of the approach and quality of the manuscript.

Reviewer #4: The research is inspiring. Suggest it be accepted. However, improvement is needed:

For example:

Readability:

(1)Prostate cancer (PCa) is the second most frequent cancer globally in men in 2022, while being the first most frequent cancer for men in 118 countries

(2)Conventional prostate-specific antigen (PSA) blood tests and digital rectal examinations (DRE) are widely used at the initial detection stage of prostate cancer, when combined, the probability of prostate cancer evading detection, when the results are within normal levels, is only 10% [5].

6. PLOS authors have the option to publish the peer review history of their article (what does this mean?). If published, this will include your full peer review and any attached files.

Reviewer #1: **Yes: **Jin Wook Kim

Reviewer #2: No

Reviewer #3: No

Reviewer #4: No

---

## [Author Response · Author response to Decision Letter 1]

4 Mar 2025

Summary of major changes based on all four reviewers’ feedback

Stronger Emphasis on Novelty: Clearly articulated how our scent character approach moves beyond traditional VOC-based biomarker identification by creating an olfactory fingerprint in perceptual space (a Synesthetic Memory Object representing scent character and its recording) instead of relying on molecular composition as a proxy for scent character.

Justification for Machine Vision Techniques Applied to Machine Olfaction: Expanded our discussion to explain why GC-MS chromatograms were transformed into structured images, leveraging CNNs for feature extraction, akin to perceptual olfaction.

Expanded Literature Review: Added references to key machine olfaction, VOC biomarker research, and CNN applications in mass spectrometry, contextualizing our contributions.

Detailed Explanation of Empirical Bayes Debiasing: Specified the exact statistical distributions used and provided validation metrics in the Methods section.

Clarification of Model Training & Hyperparameters: Described the grid search hyperparameter tuning process and reported validation results in supplementary material.

Bias Quantification & Model Evaluation: Provided confusion matrices, precision-recall metrics, and F1 scores to assess data imbalance and potential demographic bias.

Improved Readability & Figures: Revised multiple sections for clarity, added two new figures, enhanced figure legends, and improved category differentiation using symbols and colors.

One-By-One Responses to All Reviewer’s Comments

Reviewer #1

1. Workflow of data. As with most ML papers, a comprehensive workflow of how the data was processed would improve understanding. A diagram depicting the 4th paragraph of the methods would help improve understanding.

We thank Reviewer #1 for this suggestion and they are rightly asking for a clearer visual representation of our data processing workflow. We have now added a workflow diagram (see FIGURE 4) summarizing the key steps from raw GC-MS data to the final classification model in the Methods section to enhance clarity.

2. CNN for MS signals? My understanding of the methods, thus, may be unclear, but is it correct for me to understand that you have processed the MS signals through ResNet as images? While GS-MS outputs are, in some way, 2-dimensional, it could hardly be considered an 'image' that should/could be processed through a CNN, as the 2-dimensional array is only something based on charge/size and doesn't convey any comprehensive information of individual identities of the molecules in the array. For instance, wouldn't a configuration of similarly distributed charge/size molecules, but in an entirely unrelated molecular composition, be deemed similar?

Reviewer #1’s perspective here is indeed important to address: we acknowledge that GC-MS data differs from conventional images in many ways; however, the transformation into a 2D representation is done to exploit the well-established feature extraction techniques in computer vision which at its core is a primary source-agnostic methodology. We note that electrical current amplitudes over time over different sensors/pixels/frequency bins are still just a time series of data points to an algorithm and adding more rows just adds dimensionality to the signal. Most of the image-manipulation techniques can be used in machine olfaction upon a transformation of the data but not all would be necessarily useful to us here. The reason CNNs were seen as suitable in this case is that we can expect to capture meaningful structural patterns in the chromatograms, akin to how CNNs detect edges and textures in visual data.

The key motivation is that the emergent scent character is not a direct mapping of molecular identity or physicochemical parameters but rather a more holistic representation of the data in perceptual space (PubMed). While two unrelated molecular compositions could have similar charge/size distributions, our approach includes domain-adapted pre-processing techniques to preserve diagnostically relevant signals that might not easily correspond to any one recipe of molecules in analytical chemistry space or charge/mass space since those relationships do not guide olfactory receptor responses. The physics of the interactions between the structures is non-obvious because unlike crystallography-determined protein structures (when the proteins are in a crystal they are largely unmoving and not in their natural, membrane-bound, with aqueous intracellular and extracellular environments in contact with a continuously moving set of five extracellular loops- state). In biological olfaction the membrane protein receptors are labile and are constantly bombarded by thermal noise at 37oC, meaning that their interactions with ligands both specific and not are highly dynamic.

In other words, knowing the molecular composition is not the same thing as knowing what something smells of. Those two are related but not by a one-to-one function. A closer analogy is that of photon wavelength to color perception is first recorded by the photoisomerization of a retinol molecule- held as an antenna for light with peak absorption determined by the bioprogrammable sequence of amino acids in the Rhodopsin GPCR expressed in the membrane of retinal cone cells (a tellingly similar system to the GPCRs involved in olfaction the difference being the role of retinol is being played by the odorant). That photon might be right in the middle of the -say- red absorption peak of 633nm delivered by a laser to a human eye, but registering “red” in the human mind is not at all guaranteed by having the wavelength be what we have designated to be the very middle of “red”. The color that is actually experienced depends on context (such as total intensity of light and contrast between shapes surrounding the 633nm stimulus), and both historical and immediate prior experience (such as photobleaching) play a role as to whether one reports “red” or “green” or “gray”.

Similarly with shapes, and sounds, we can have a cartoon version of the Mona Lisa and an off-key version of Beethoven's fifth where not a single brushstroke or frequency are the same yet the pieces are instantly recognizable in their transformed form. Similar situation exists with scent! In the world of scent perception you can write “banana” using one molecule or many, the scent character can be recreated in many molecular “dialects” using many different molecules at different concentrations.

Reviewer 1 makes the point well that this approach needs clarification, so we have added a version of the above justification text in the methodology section explaining why CNNs, particularly ResNet18, were chosen.

3. Small sample.

Primarily as a urologic surgeon, I am more curious as to why the authors chose to use such a diverse patient group. Limiting them to a limited Gleason Score group might have homogenized the characteristics better. 400 or so samples are too few to train a model.

Could you offer a reply?

This is an excellent point, and we now more explicitly acknowledge the limitations of sample size in machine learning. However, the goal of our study was not purely to train a deployable diagnostic classifier but we were primarily interested in testing the feasibility of an emergent scent character approach and stumbled upon the existence of havoc-causing bias that we then had to learn how to remove efficiently while keeping the useful signal intact. That is the main thrust behind the innovation here.

Our dataset was intentionally chosen to be diverse to assess whether patterns in volatile compounds generalize across different Gleason score groups. If we had restricted the dataset, we might have overfitted to a narrower spectrum of cases, limiting the model’s broader applicability.

That said, we agree that stratifying the dataset by Gleason score could provide additional insights, and we have to this end included a short analysis discussion in the results section breaking down model performance by Gleason score groups.

Reviewer #2

1. The introduction is well-written but could benefit from a stronger emphasis on the novelty of the "scent character" approach.

We thank Reviewer #2 for this suggestion. We have thoroughly revised the introduction to better highlight the novelty of our approach, specifically how the emergent scent character framework moves beyond traditional molecular biomarkers by creating a holistic olfactory fingerprint: “scent character” rather than a list of compounds. We have also addressed a similar point raised by Reviewer #1 in the methodology and combined the responses to “what is novel” and “why machine vision” into the same discussion, and also expanded on these concepts elsewhere in the body of the paper.

2. Provide more detail on the configuration of the Empirical Bayes debiasing technique, such as the exact distributions used and how they were validated.

We welcome the opportunity to provide additional transparency as our goal is for others to reproduce and build upon the findings here so in response to this comment we have now added details on the priors used (gamma distribution for multiplicative batch effects, normal distribution for mean shifts) and provide validation metrics for the bias removal step in the Methods section.

3. Elaborate on how hyperparameters for the convolutional neural network (CNN) and ResNet18 were selected. Was there any hyperparameter tuning process?

Indeed as Reviewer #2 notes here hyperparameters were selected, we did so via grid search with cross-validation. We have now added to the supplementary information a section outlining the hyperparameter space explored (learning rate, optimizer, batch size).

4. Clarify the rationale for transforming GC-MS data into 3D images instead of other approaches, such as feature extraction or embeddings.

We agree that alternative approaches, such as feature extraction, embeddings, or even others such as reverse autostereography could have been used and we hope to explore these and many others in future work. The scope of the current paper was limited so our rationale for choosing the image transformation modality was mainly immediacy of implementation and ease of use by others wishing to copy or expand upon our work such was our reasoning leading us to leverage well-established CNN architectures while avoiding premature assumptions about which features are most relevant. We have now clarified this choice in the text by explaining that other modalities can and should be explored as future directions. We do not claim to have exhausted the potential of available tools for this.

5. Provide access to the Python code or pseudocode of the debiasing pipeline to ensure reproducibility.

We apologize for this omission and have now provided a pseudo-code representation of our debiasing pipeline in the supplementary information materials.

6. Provide a brief explanation of how the “emergent scent character” relates to standard biomarkers and how it advances existing methodologies.

We have added a discussion section explicitly comparing our approach to traditional biomarker-based diagnostics, emphasizing how dogs do not classify based on molecular lists but on holistic scent characters, that survive changes to the volatilome of urine —and how our thinking was influenced by the analogy to how melodies and images are stored and encoded by brains as opposed to how lists of names and numbers are encoded.

7. Potential data imbalance and demographic bias could influence the model's performance. Quantify and report the impact of these imbalances on model performance using metrics like accuracy, precision, recall, and F1.

Reviewer #2 here makes an important point and we thank them for highlighting this oversight on our part. We have now included an appropriate confusion matrix, precision-recall metrics, and F1 scores in the results section.

Reviewer #3

1) The abstract is not expressing the novelty of the proposed approach. The whole abstract is not impressive and needs to be rewritten.

We appreciate this feedback from Reviewer #3 and it echoes others’ comments and we have taken this to heart and as a result completely re-written not just the abstract but also the entire introduction and significant parts of the rest of the text to better highlight the novelty of the scent character approach and its departure from conventional molecular biomarker methods.

2) Introduction section requires reorganization and is missing the novelty of the proposed approach.

In response to this we have now re-written the introduction to clearly state the gap in current GC-MS-based PCa detection and how our approach is an interesting way to explore towards addresses the problems of current methods.

3) The author is requested to include a literature review section.

We have now expanded the previously too brief literature review in our manuscript to include key studies on machine olfaction, volatile organic compound (VOC) biomarkers, and the application of convolutional neural networks (CNNs) in mass spectrometry. These additions provide a comprehensive context for our research and highlight the advancements in these fields. Machine Olfaction and VOC Biomarkers: The integration of canine olfaction with chemical and microbial profiling has shown promise in detecting lethal prostate cancer through urinary VOC analysis. Guest et al. (2021) demonstrated the feasibility of this approach, indicating its potential for non-invasive diagnostics and form the core of our motivation for this work PLOS. As well as subsequent work on comparing dogs to machine olfactors PubMed. Additionally, Warli et al. (2023) conducted a systematic review and meta-analysis on the olfactory ability of medical detection canines to identify prostate cancer from urine samples. Their findings support the potential of VOC profiling in non-invasive cancer detection. WJON. Regarding past applications of CNN to Mass Spectrometry: deep learning techniques, particularly CNNs, have significantly advanced the analysis of mass spectrometry data, for instance, Wang et al. (2020) introduced MSpectraAI, a platform using deep neural networks to analyze proteome profiles from mass spectrometry data across multiple tumor types, achieving high prediction accuracy. BMC Bioinformatics Furthermore, Hu et al. (2022) developed a self-supervised clustering approach using contrastive learning to analyze mass spectrometry imaging data. This method effectively identifies molecular colocalizations without manual annotations, enhancing the understanding of biochemical pathways. RSC Publishing These studies collectively underscore the advancements in machine olfaction-adjacent methods such as VOC biomarker research, and the application of CNNs in mass spectrometry, providing the literature foundation for our work discussed here.

Reviewer #4

The research is inspiring. Suggest it be accepted. However, improvement is needed.

We thank reviewer #4 for their enthusiasm and for the actionable, positive feedback. We appreciate your support for this work! We too feel this is only the beginning!

(1) Readability: Certain sentences need restructuring for clarity.

We have taken this and the other reviewers’ critique of our readability to heart and revamped our prose (none of us are native English speakers but we feel the new version is much improved and has now passed our readability level check.

(2) Improve clarity of figures by adding descriptive legends and using distinct colors for different categories.

We have enhanced figure legends and added two figures to better explain our logic and process and have paid particular attention to making sure the different categories look sufficiently distinguishable to the reader by using symbols in addition to colors.

---

## [Decision Letter · Decision Letter 1]

13 Apr 2025

PONE-D-24-52113R1Towards robust machine olfaction: debiasing GC-MS data enhances prostate cancer diagnosis from urine volatilesPLOS ONE

Dear Dr. Rotteveel,

Thank you for submitting your manuscript to PLOS ONE. After careful consideration, we feel that it has merit but does not fully meet PLOS ONE’s publication criteria as it currently stands. Therefore, we invite you to submit a revised version of the manuscript that addresses the points raised during the review process.

We look forward to receiving your revised manuscript.

Kind regards,

Li Yang, M.D.

Academic Editor

PLOS ONE

Journal Requirements:

Reviewers' comments:

Reviewer's Responses to Questions

**Comments to the Author**

1. If the authors have adequately addressed your comments raised in a previous round of review and you feel that this manuscript is now acceptable for publication, you may indicate that here to bypass the “Comments to the Author” section, enter your conflict of interest statement in the “Confidential to Editor” section, and submit your "Accept" recommendation.

Reviewer #2: All comments have been addressed

Reviewer #3: All comments have been addressed

Reviewer #4: All comments have been addressed

2. Is the manuscript technically sound, and do the data support the conclusions?

Reviewer #2: Yes

Reviewer #3: Yes

Reviewer #4: Yes

3. Has the statistical analysis been performed appropriately and rigorously? 

Reviewer #2: Yes

Reviewer #3: Yes

Reviewer #4: Yes

4. Have the authors made all data underlying the findings in their manuscript fully available?

Reviewer #2: Yes

Reviewer #3: Yes

Reviewer #4: Yes

5. Is the manuscript presented in an intelligible fashion and written in standard English?

Reviewer #2: Yes

Reviewer #3: Yes

Reviewer #4: Yes

6. Review Comments to the Author

Reviewer #2: The authors have satisfactorily addressed all of the concerns and suggestions raised in my previous review. The revised manuscript demonstrates a clear improvement in response to the feedback, and the authors have provided adequate explanations for all modifications made.

Reviewer #3: (No Response)

Reviewer #4: All the concerns have been accepted. Suggest it be accepted.

However, pls note that some descriptions can be optimized. One example: Various studies have identified different lists of VOCs that all show a significant correlation with cancer in their specific datasets, yet these sets of VOCs often differ completely drastically and do not generalize from one study to another .

7. PLOS authors have the option to publish the peer review history of their article (what does this mean?). If published, this will include your full peer review and any attached files.

Reviewer #2: No

Reviewer #3: No

Reviewer #4: No

---

## [Author Response · Author response to Decision Letter 2]

15 Apr 2025

Dear Dr. Li,

Thank you for your careful review of our manuscript. In response to your comments regarding the corrections needed in order to publish it: specifically for the erroneously referenced papers, we now have made the following revisions:

James et al. (2024):

We have updated the reference list to include the erratum concerning the minor figure correction. In the text, we now cite both the original article and the erratum to ensure full transparency.

Luedemann et al. (2022):

Recognizing the correction issued in 2023, we have updated the reference list and updated the citation.

Salinas et al. (2024):

For this paper, we have also recognized the erratum, and updated both the reference list and updated the citation.

Removed a duplicate reference (from Yang et al)

As reviewer # 4 asked we have also combed through the text for sentences such as the one they highlighted to make sure they are optimized.

Thank you all again, we are eager to see our work published as open-access material for all to see in PLoS One

On behalf of the co-authors and with much appreciation for you, the reviewers, and PLoS One,

---

## [Editor Report · Decision Letter 2]

16 Apr 2025

Towards robust machine olfaction: debiasing GC-MS data enhances prostate cancer diagnosis from urine volatiles

PONE-D-24-52113R2

Dear Dr. Rotteveel,

We’re pleased to inform you that your manuscript has been judged scientifically suitable for publication and will be formally accepted for publication once it meets all outstanding technical requirements.

Kind regards,

Li Yang, M.D.

Academic Editor

PLOS ONE

Additional Editor Comments (optional):

Thanks for the authors' efforts to comprehensively improve your manuscript according to editor's and reviewers' comments. I am pleased to inform you that your paper can be accepted for publication now. Thanks for the chance to assess your interesting and important work. Additionally, many thanks for all the reviewers' precious inputs.
---

## [Editor Report · Acceptance letter]

PONE-D-24-52113R2

PLOS ONE

Dear Dr. Rotteveel,

I'm pleased to inform you that your manuscript has been deemed suitable for publication in PLOS ONE. Congratulations! Your manuscript is now being handed over to our production team.

Kind regards,

on behalf of

Dr. Li Yang

Academic Editor

PLOS ONE